# Is mHealth a Useful Tool for Self-Assessment and Rehabilitation of People with Multiple Sclerosis? A Systematic Review

**DOI:** 10.3390/brainsci11091187

**Published:** 2021-09-09

**Authors:** Bruno Bonnechère, Aki Rintala, Annemie Spooren, Ilse Lamers, Peter Feys

**Affiliations:** 1REVAL-Rehabilitation Research Center, Faculty of Rehabilitation Sciences, Hasselt University, B-3590 Diepenbeek, Belgium; annemie.spooren@uhasselt.be (A.S.); ilse.lamers@uhasselt.be (I.L.); peter.feys@uhasselt.be (P.F.); 2Faculty of Social Services and Health Care, LAB University of Applied Sciences, FI-15210 Lahti, Finland; aki.rintala@lab.fi; 3University MS Center Hasselt-Pelt, B-3500 Hasselt, Belgium

**Keywords:** mHealth, multiple sclerosis, telemonitoring, longitudinal assessment, rehabilitation, fatigue, walking, cognition

## Abstract

The development of mobile technology and mobile Internet offers new possibilities in rehabilitation and clinical assessment in a longitudinal perspective for multiple sclerosis management. However, because the mobile health applications (mHealth) have only been developed recently, the level of evidence supporting the use of mHealth in people with multiple sclerosis (pwMS) is currently unclear. Therefore, this review aims to list and describe the different mHealth available for rehabilitation and self-assessment of pwMS and to define the level of evidence supporting these interventions for functioning problems categorized within the International Classification of Functioning, Disability and Health (ICF). In total, 36 studies, performed with 22 different mHealth, were included in this review, 30 about rehabilitation and six for self-assessment, representing 3091 patients. For rehabilitation, most of the studies were focusing on cognitive function and fatigue. Concerning the efficacy, we found a small but significant effect of the use of mHealth for cognitive training (Standardized Mean Difference (SMD) = 0.28 [0.12; 0.45]) and moderate effect for fatigue (SMD = 0.61 [0.47; 0.76]). mHealth is a promising tool in pwMS but more studies are needed to validate these solutions in the other ICF categories. More replications studies are also needed as most of the mHealth have only been assessed in one single study.

## 1. Introduction

People with Multiple Sclerosis (pwMS) may manifest heterogeneous symptoms and functioning problems that require continuous and long-term rehabilitation programs in clinical and community settings across the disability spectrum. In high-income countries, the pressure on healthcare systems is increasing [1] and the continuity of high-level care is threatened due to lack of reimbursement, while in some countries access to the specialized MS centers has always been poor [2]. Furthermore, a vast majority of pwMS often present fatigue, emotional or cognitive dysfunction, or restricted physical mobility or a combination of those which limits access to rehabilitation centers. In this context, the WHO stated that lack of access to specialized centers or healthcare professionals is one of the most important limitations for the rehabilitation process [3]. The use of mobile technologies and electronic health (eHealth) could be an alternative to tackle the above-mentioned limitations (i.e., lack of access to centers) of rehabilitation of pwMS or complement current rehabilitation services. eHealth is also expected to facilitate the monitoring of functioning of pwMS between medical consultations, which is informative to define whether to continue or adapt medical treatment.

The number of healthcare interventions delivered via personal mobile devices (mHealth) has increased exponentially thanks to the availability of mobile technology (the number of smartphone subscriptions worldwide today surpasses six billion and is forecast to further grow by several hundred million in the next few years [4]). The development and implementation of mHealth open new perspectives and opportunities in the healthcare sector. Previous studies highlighted that mHealth has already been accepted by patients. Amongst the most important benefits identified by the patients are easy access to personalized information, convenience, better information on their health, and the ability to communicate more easily with healthcare professionals [5,6]. So far, most studies have focused on patients with cancer [7,8,9], patients with cardiovascular diseases [10,11], or older adults with [12] or without cognitive impairment [13].

Concerning pwMS, a meta-analysis showed that technology-based distance physical rehabilitation intervention has a positive effect on physical activity and walking ability when compared to usual care or no intervention [14]. Another review synthesized the different eHealth technologies that are available for the management of pwMS [15]. eHealth is a broader term than mHealth; eHealth is composed of the electronic records, self-remote disease monitoring (i.e., blood markers, vital signs), mobile and wired communication for advice and education, and tools to facilitate self-management (i.e., physical activity tracker, rehabilitation exercises reminder or calendar). This previous review was published in 2018 and given the important development of technology-supported rehabilitation tools a lot of new solutions have been developed. Furthermore, there is currently a lack of information about the different mHealth currently available, and the level of evidence supporting them, in pwMS.

Therefore, the aim of this paper is first to describe the different mHealth applications currently available to assist the rehabilitation of pwMS and the tools that exist to perform longitudinal self-assessment of the patients. The second aim of this review is to determine the level of evidence supporting the use of mHealth in pwMS on their functioning according to the International Classification of Functioning, Disability, and Health (ICF).

## 2. Methods

### 2.1. Search Strategy and Selection Criteria

Records were searched on three databases (Pubmed, Biber, and Scopus) to identify eligible studies published between 2011 (after the release of the first generation of iPad, which was an important step in the development of mobile applications) and June 2021. MeSH terms and free words referring to e-health intervention in pwMS (‘multiple sclerosis’, ‘ms’, ‘ehealth’, ‘mhealth’, ‘mobile apps, smartphone intervention’, ‘apps’, ‘self-monitoring, ‘self-assessment’, ‘functioning’, ‘intervention’, ‘rehabilitation’) were used as keywords. The complete search strategy is presented in Appendix A.

### 2.2. Eligibility Criteria

A PICOs approach was used as inclusion and exclusion criteria, which were assessed by the study team [16].
**Population**: pwMS performing training (rehabilitation exercises) or self-assessment in home-environment, studies with inpatient treatment or assisted-rehabilitation were not included.**Intervention**: mHealth rehabilitation intervention (planned and supervised interventions), or self-assessment studies with repeated measurements over time, using any type of support (e.g., smartphones, phones, apps, web applications). Studies using non-specific games, virtual reality or active video games (e.g., Nintendo Wii, Microsoft Xbox Kinect), or computer-supported therapy were not included.**Control**: usual care or no intervention.**Outcome measures**: any type of outcome measure related to the International Classification of Functioning, Disability and Health (ICF).**Study design**: RCTs, explorative studies.

A flow diagram of the study selection with the screened articles and the selection process is presented in Figure 1.

### 2.3. Quality Assessment

Since we included different types of articles, the critical appraisal of the methodological quality was based on the Downs and Black checklist [17], as this checklist is the best option to assess the quality and risk of bias for both RCT and non-RCT [18].

### 2.4. Data Extraction

The following information was extracted from the included studies: characteristics of the patients (age, sex ratio, type of MS and severity), type and duration of the mHealth intervention, study design, main outcomes, and ICF domains evaluated.

### 2.5. Statistical Analysis

For studies assessing the efficacy of a rehabilitation program, we performed a meta-analysis. The measure of treatment effect was the standardized mean difference effect size (standardized mean difference (SMD)), defined as the between-group difference in mean values divided by the pooled SD computed using the Hedge’s g method ($Hedges’g=\frac{M_{1}-M_{2}}{SD_{pooled}}$). If different tests were used to assess the same ICF domains in the same study, the different results were pooled to have one unique SMD as recommended by Cochrane’s group [19]. A positive SMD implies better therapeutic effects in the intervention group compared to the control. We assessed the heterogeneity in stratified analyses by type of ICF domains. We calculated the variance estimate tau^2^ as a measure of between-trial heterogeneity. We prespecified a tau^2^ of 0.0 to represent no heterogeneity, 0.0–0.2 to represent low heterogeneity, 0.2–0.4 to represent moderate heterogeneity, and above 0.4 to represent high heterogeneity between trials [20]. To deal with high or moderate heterogeneity we used random-effect models and presented forest plots for the different ICF domains. We checked for publication bias using funnel plot [21] and Egger’s test for the intercept was applied to check the asymmetry [22].

### 2.6. Ethical Approval

This systematic review was reported following the Preferred Reporting Items for Systematic Reviews and Meta-Analyses (PRISMA) recommendations [23]. For the present study, no ethics committee approval was necessary.

## 3. Results

For the sake of clarity, this section has been divided into three different parts; first, we will present the characteristics of the included studies and the patients; then we will describe the different mHealth used in these studies and finally, we will present the clinical efficacy for the different domains in the ICF.

### 3.1. Search Results

In total, 1346 articles were found with the systematic review. A total of 112 full-text articles were assessed and 36 papers were included in the analysis. The PRISMA flowchart on the study selection is presented in Figure 1.

### 3.2. Characteristics of the Included Studies

Thirty studies about the use of mHealth for rehabilitation interventions of pwMS were included in this review, representing a total of 1962 patients [24,25,26,27,28,29,30,31,32,33,34,35,36,37,38,39,40,41,42,43,44,45,46,47,48,49,50,51,52,53]. The majority of these studies (*n* = 25; 3%) were RCTs. Concerning the patients, the majority of the patients were female (76 ± 10%); concerning the type of MS the majority of the included patients (79%) have relapsing-remitting multiple sclerosis (RRMS), 16% have secondary progressive multiple sclerosis (SPMS) and 5% primary progressive multiple sclerosis (PPMS), and the average EDSS is 3.5 ± 1.1. The median duration of the intervention was 9 weeks [p25 = 8 weeks, p75 = 12 weeks] for a median time of 18 h [p25 = 13.25 h, p75 = 27 h]. Finally, for the ICF, 16 (53%) of the studies reported outcomes related to cognition, 11 (37%) to fatigue, 10 (33%) to quality of life, 7 (23%) on motor function, and 6 (20%) on activity level; we observed that most of the studies are assessing different primary outcomes (ICF domains). The complete description of the included studies is presented in Table 1. Amongst the 30 studies, 16 different mHealth apps have been tested.

Concerning the self-assessment tools six studies, using six different mHealth applications, were included in the review, representing 1129 participants (955 pwMS [88% with RRMS, 5% with SPMS and 7% with PPMS, average EDSS 2.5 ± 0.5] and 174 healthy participants) [54,55,56,57,58,59]. The median duration of the follow-up was 12 weeks [p25 = 6 weeks, p75 = 24 weeks].

The characteristics of the studies and participants are summarized in Table 2.

### 3.3. Quality Assessment

The quality of the included papers was checked using the Downs and Black checklist. Overall, the average score for the included studies is 21.9 out of 28 (22.2 for studies on rehabilitation, 20.3 for studies assessing self-assessment). The average results for the different questions and sub-analysis of the Downs and Black checklist are presented in Figure 2.

When analyzing individual items, we observed that, due to the nature of the training, the blinding of the participants was not possible, on the other hand, the blinding of the investigators was not guaranteed either. Another potential source of bias is the uncertainty about the randomization assignment until the complete recruitment. The last important point is that a high number of studies do not take into consideration the patients that did not complete the intervention (loss in follow-up) so leading to uncertainty on reasons of non-adherence. Only a few studies used intention-to-treat analysis. On average 90.6% of the included patients completed the entire protocol and were included in the final analysis.

### 3.4. Description of the Available mHealth Solutions

First, we present the mHealth solutions that are mainly used for rehabilitation purposes. Most of the proposed mHealth solutions have been studied for cognition, QoL, and fatigue and were limited to one single ICF domain. We later discuss the applications for the respective domains, although some overlap occurred.

RehaCom [24,30,33,48] is a comprehensive and sophisticated system of software for computer-assisted cognitive rehabilitation. It proposes different solutions for screening and training cognitive functions and offers apps for home training.

BrainHQ [32,41,42] is a platform providing a set of more than 30 mini brain training exercises designed to challenge different cognitive functions (processing speed, attention, working memory, and executive function through visual and auditory domains). The initial level of challenge is low and the difficulty is adapted on an individual basis as learning and abilities improve over time. The company was previously known as Posit Sciences [27].

Luminosity [25] is a platform providing cognitive training exercises embedded in games. As for BrainHQ different cognitive functions can be challenged in a set of different mini-games.

The Memory, Attention, and Problem Solving Skills for Persons with Multiple Sclerosis (MAPSS-MS) intervention [35] aims to help pwMS acquire the highest level of cognitive functioning and functional independence. It includes problems solving and lifestyle adjustments (sleep, stress management, physical activity) that support cognitive functioning and will support persons with MS in the use of compensatory cognitive strategies and cognitive skills. The cognitive training is done with Luminosity app.

BrainStim [28] is a computerized training tool based on the working memory (WM) model of Baddeley [60]. It consists of three different modules targeting both, verbal and visual-spatial aspects of WM.

COGNI-TRAcK [31] implements three different types of exercises which were shown to be effective in improving the cognitive status of healthy subjects. The exercises consisted of (i) a visuospatial WM task; (ii) an “operation” N-back task; (iii) a “dual” N-back task [61].

The Attention Processing training (APT) program [25] is a cognitive rehabilitation intervention that targets focused, sustained, selective, alternating, and divided attention. The aim is to increase the ability to respond to specific stimuli [62].

ELEVEDIA [37] content is based on cognitive behavioral therapy strategies and is conveyed chiefly via the technique of a ‘simulated dialogue’. Program modules are composed of an introduction and a summary and include homework tasks. Patients are advised to access the program once to twice per week.

The MS Home Automated Telehealth (MS HAT) system [34,43] is supporting patient-centered care, self-management and allows easier patient–provider communication. Three interfaces are available: patient unit, server, and clinical unit. The patient unit had interactive options for data collection, educational content, exercise information, and therapist–patient communication, access to exercises, response to exercise-specific assessments, and documenting exercise data from home. Exercises consist of task-oriented training such as digitized writing tracking or manipulating light bulbs or keys. Exercise adherence feedback was via diary entries, calendars, and graphs [63].

AKL-TO3 [51] is engaging the patients in simultaneous sensory and motor tasks and is designed to engage the frontal neural network. It enabled real-time monitoring of progress and continuously challenges patients so that the training is never too easy or difficult encouraging patients to improve performance.

RELAXaHEAD [49] is designed for pain management and in particular migraine and neck pain. It contained a headache diary, which includes features for tracking headache characteristics, headache medications and sleep, and tracking medication side effects and menstrual cycles. The app also contains a serious game module to ease muscle relaxation.

WalkWithMe [50] has been developed to motivate and stimulate patients to walk more and farther. It allows to track the walking activities and follow up on progress. The app detects the walking speed and gives feedback during walking with verbal feedback (i.e., pace) by the virtual coach.

Webbasedphysio.com [47] is an internet-delivered therapeutic exercise program. The web-based physio allows people the flexibility to do their own, individualized exercise program at a time and location which is convenient to them, thus enhancing the individual’s ability to self-manage their condition on a long-term basis.

Deprexis [29] is an online program based on principles of cognitive-behavioral therapy. It consists of 10 sequential modules—psycho-education, behavioral activation, cognitive modification, mindfulness and acceptance, interpersonal skills, relaxation, physical exercise and lifestyle modification, problem-solving, expressive writing and forgiveness, positive psychology, and emotion-focus interventions.

The Mindfulness Based Stress Reduction (MBSR) [38] deals with stress management, relaxation training, sleep hygiene, fatigue, and social relationships. The course materials were developed using existing informative MS videos, created by the Italian MS Association, recording new interviews and generating new exercises.

Concerning the mHealth apps that are mainly used for self-assessment:

MSdialog [55] is a web and mobile (i.e., cell phone and tablet) based software application that combines information from RebiSmart (with health information recorded by patients via their personal computer or smartphone to collect and store real-time data regarding administration of Rebif (interferon β-1a), clinical outcomes, and patient reported outcomes). MSdialog offers a practical means by which patients record and exchange information with their healthcare specialists intending to support the patient–physician relationship and offering patients a method of engaging in the pharmaceutical management of their MS and patients’ self-reported outcomes [64].

MS Telecoach [56] provides a combination of monitoring, self-management, and motivational messages, focusing on energy management of physical activity to improve fatigue levels. It has two components: telemonitoring (physical activity through accelerometers and self-reported fatigue impact levels) and tele-coaching (motivational messages and advice).

Floodlight [57] is a combination of continuous sensor data capture and standard clinical outcome measures. It involves performing a set of daily active tests to evaluate cognition, upper extremity function, gait, and balance domains and contribute sensor data via passive monitoring, also including self-reported patient outcomes.

The Mellen Center Care Online (MCCO-enhanced) [54] is an electronic messaging system between clinician and patient. It contains a self-monitoring and self-management system to assess MS symptoms and the pwMS receives graphical feedback to evaluate symptom changes

FatigueApp.com [58] is collecting data to correlate fatigue measures with other symptoms and quality of life. Fatigue questionnaires are completed every morning for 6 consecutive days and again 4 weeks later.

ElevateMS [59] allows collecting different data in the real-world environment of the patients such as self-reported measures of symptoms and health via ‘check-in’-surveys Independent assessments of motor function occur via sensor-based active functional tests, participants are encouraged to complete surveys daily, and notifications to perform more comprehensive functional tests are provided once a week.

### 3.5. Outcome Data Related to ICF

#### 3.5.1. Rehabilitation

Amongst the included RCTs, 20 were included in the meta-analysis assessing the efficacy of mHeath for rehabilitation [24,26,27,28,29,30,31,33,34,35,36,37,39,40,44,45,47,48,49,51,65], representing 1393 pwMS. When considering all the studies and ICF domains together, the heterogeneity between the studies was moderate (tau^2^ = 0.30, 95%CI [0.26; 0.62]), therefore we decided to use random-effect model. The funnel plot did not show significant asymmetry (Egger’s intercept = 0.45, *p* = 0.91) (Appendix A). The sensitivity analysis did not show any outlier (Appendix A).

The overall effect of mHealth intervention in pwMS is moderate (SMD = 0.50 [0.35; 0.66]) and statistically significant (*p* < 0.0001). Since different studies evaluated human functioning at different aspects according to the ICF, we then performed subgroup analysis to assess the efficacy across the different ICF. The forest plot is presented in Figure 3.

At the ICF domains level, we observed the biggest effect for fatigue (SMD = 0.61 [0.47; 0.76]), followed by outcome measures at the activity level (SMD = 0.56 [0.25; 0.87]) and cognitive impairment (SMD = 0.28 [0.12; 0.45]). Note that for activity level these results must be interpreted carefully due to the small number of included studies (*n* = 3). For the domains of motor function and quality of life the results were not significant but only included two and three studies, respectively. Using a fixed-effect model to summarize the overall ICF functioning we found an overall moderate effect (SMD 0.47 [0.37; 0.56], *p* < 0.0001).

We then summarized the main results and conclusions of the studies that were not included in the meta-analysis.

Concerning cognitive function, Fuchs et al., 2019 investigated the clinical characteristics predicting response to a home-based restorative cognitive training. Significant improvements were observed after training [41]. Villou et al., 2020 is an explorative study that reported statistical improvement of various cognitive functions after training such as visuospatial memory, visual attention, task-switching, reading speed and response inhibition, and verbal learning [42].

For fatigue, Stuifenbergen et al., 2018 analyzed the acceptability and effect of MAPSS-MS on cognitive function and fatigue. The authors find similar results as with usual care; interestingly, the improvements were maintained during the follow-up at 3 and 6 months and were superior in the intervention group [35].

For the quality of life, Cavalera et al., 2018 showed an improvement of QoL after 8 weeks of intervention using a mindfulness program but the progress was not maintained over time (6-month follow-up after the end of the intervention) [38]. Tarakci et al., 2021 compared an in-person rehabilitation program with a telerehabilitation program. After 12 weeks of training, the results were similar in the two groups for fatigue and activity level [52]. Manns et al., 2020 demonstrated a reduction of fatigue after a combined intervention (SitLess and MoveMore) but the difference was not significant compared to usual care [46]. Interestingly the total sedentary time decreased in the intervention group and these results are maintained over time.

Van Geel et al. reported that using the WalkWithMe app induced a significant improvement in quality of life, walking, and leisure, 36-Item Short-Form Health Survey (SF-36) quality of life, cognition, cognitive fatigability, lower limb strength, and dominant hand function. However, it was an observational study without a control group [50].

#### 3.5.2. Self-Assessment

Concerning the efficacy of self-assessment and monitoring, only six studies were included in this review.

Miller et al. highlighted group differences between the MCCO-original and MCCO-enhanced groups. MCCO-original had a higher European Quality of Life level after 12 months of regularly self-monitoring their quality of life [54].

Greiner et al. performed a 6-week longitudinal observation and showed that MSdialog was adapted to monitor patient-reported outcomes. Amongst the different functions evaluated by the pwMS, fatigue (99%), physical health (96%), cognitive deficits (93%), pain (91%) and sleep quality (91%) were the most important. These numbers represent the weight given by the patients for these different functions that scored the MS quality-of-life questionnaire using a visual analogical scale [55].

D’Hooghe et al. showed that it is feasible to use the MS TeleCoach at home without supervision. The authors observed a significant decrease in fatigue and an increase in cognitive function after 12 weeks of use [56].

Midaglia et al. assessed the usability and acceptability of the Floodlight for active monitoring and passive monitoring intervention. After 24 weeks of intervention, mHealth had an acceptable impact on daily activities including cognition and physical activity for 80% of the pwMS [57].

Newland et al. reported that the FatigueApp could collect self-reported symptoms including fatigue, self-reported EDSS (EDSS-SR), pain, and cognition [58]. Participants were asked to complete the questionnaires for 7 consecutive days and then again 4 weeks later.

Pratap et al. in a large study including more than 500 pwMS described that ElevateMS can be used to longitudinally (12 weeks period) to collect information about the most common symptoms of MS. During this follow-up, they observed that the most frequent complaints are fatigue (63%), memory issues (42%) and difficulty with walking (41%). After the intervention, there were significantly increased functional performances and QoL [59].

### 3.6. Summary

To summarize the findings of this study we listed the different mHealth according to the targeted ICF domain for both rehabilitation and self-assessment in Table 3.

## 4. Discussion

The main result of this review is the high number of solutions (applications) currently being tested with pwMS for rehabilitation (*n* = 16), despite the relatively recent development and use of these new apps in rehabilitation. On another side, the development of mHealth for self-assessment and home-monitoring is still limited (six apps found). Consequently, one of the downsides is that there are only very few studies performed with the same mHealth which makes it more difficult to compare the studies and thus to determine the level of evidence. Therefore, rather than comparing the efficacy of each particular mHealth, we performed the analysis at the ICF domain level. The most studied ICF domain is cognition: we found a small but significant effect of the training using mHealth (SMD = 0.28 [0.12; 0.45]) which is consistent with other meta-analyses summarizing the effect of computerized cognitive training, including computer solutions and supervised training (SMD = 0.30 [0.18; 0.43]) [66]. It is important to note here that there is currently a lack of information about the transfer of the benefits gained in the mHealth solution in the activity of daily living as most of the studies only assess direct or near transfer effects. The second most studied function is fatigue, with a moderate effect (SMD = 0.61 [0.47; 0.76]). The effect of mHealth is a bit lower than the effect of pharmacological treatment (i.e., amantadine): SMD = 1.09 [0.87; 1.30] [67], but similar to the effect of exercise therapy (SMD = 0.53 [0.33; 0.73]) [68].

For the motor function and quality of life, there are, currently, not enough studies available, but the few studies available also seem to indicate a favorable effect. The paucity of studies investigating the effects of mHealth applications to train motor functions is somehow surprising. However, we excluded studies including wearables and thus the number of interventions done to increase physical activity based on step count (i.e., [69]). The low numbers of studies investigating the effects of mHealth interventions on quality of life may be expected as the quality of life is often thought to be the result of improving specific ICF domains.

Another major finding of this systematic review regarding self-assessment is the fact that mHealth can be used directly by the patients to continuously monitor several different functions in their living environment. The solutions are not only well accepted by the patients, but several studies also show that using this type of mHealth is directly beneficial for the patients. This positive effect may be mediated by a better knowledge of the diseases and symptoms (education) [70] but also by the more active participation of the patient in his treatment (patients’ engagement) [71].

There are several limitations to this review. The first one is the lack of standardization in the nomenclature used to describe the different mHealth currently tested in research. Therefore, due to the small numbers of studies published, we ended up including studies assessing different types of applications and intervention modalities or duration. The heterogeneity between the studies, and the patients, makes it more difficult to compare studies and especially to generalize the results. There is also a huge heterogeneity in the duration of the intervention for both the duration of the intervention (ranging from minimum 4 to maximal 26 weeks) and the total amount of training (ranging from 4 to 65 h). Unfortunately, due to the small number of studies included in the different ICF levels, we could not perform meta-regression to determine if there is a dose–response relationship between the amount of training using the mHealth and the clinical outcome.

A third important limitation is that most of the included studies on the rehabilitation aspects (except [32,35,38,40,45]) have relatively small sample sizes and the results are likely to be underpowered [72]. Furthermore, the percentage of participants that were included in the final analysis is 90% and information about the adherence to the intervention was missing (usually a threshold of 80% is applied to determine if the participants do a sufficient amount of exercises [73]). Concerning the meta-analysis, due to the relatively small number of included studies, the results must also be interpreted carefully, especially for the ICF motor function and quality of life. Concerning motor functions, most of the current solutions are focusing on walking while patients may also experience severe disability in upper limbs functions and dexterity, efforts must be made to develop solutions that focus on these problems. Concerning the external validity of this review and the translation to the clinic, it is important to note that the vast majority of the applications were tested in pwMS with mild disability (EDSS = 3.5 ± 1.1) with RRMS (79%), and thus not guaranteed to be applicable to the same extent in more disabled patients with restricted mobility. Further studies must therefore focus on more disabled patients to determine the feasibility of mHealth with these patients if the efficacy is similar.

Finally, most of presented solutions are still at the research project stage and applications are not yet widely available to patients or their treating clinicians.

Despite these limitations, this review highlights interesting and promising results for patients. However, there are still a few points that should be addressed before these solutions can be used in daily practice. The first, and probably most important is the recognition of the m- and eHealth apps as medical devices. In June 2020, the US Food and Drug Administration (FDA) permitted the marketing of the first game-based digital therapeutic device to improve attention function in children with attention deficit hyperactivity disorder (ADHD). The mHeath, EndeavorRx, is indicated to improve attention function as measured by computer-based testing and is the first digital therapeutic intended to improve symptoms associated with ADHD, as well as the first game-based therapeutic granted marketing authorization by the FDA for any type of condition. The device is intended for use as part of a therapeutic program that may include clinician-directed therapy, medication, and/or educational programs, which further address symptoms of the disorder [74]. Interestingly this solution is developed by Akili, the company that has developed AKL-T03 which also shows significant results in pwMS [51]. The COVID-19 pandemic has not only disrupted healthcare systems but has also allowed for a very significant acceleration in the development, implementation, and recognition of mHealth in the clinics [75]. It is important to note, however, that most of the measures taken during the crisis may be temporary and it is hoped that efforts will continue in this direction once the crisis is over. For example, it will be important to adapt the nomenclature of interventions, as mobile solutions are currently placed in the same categories as drugs, which poses problems for the validation and reimbursement of these interventions [76]. Another limitation is that, for the moment, the majority of the analyzed mHealth is being developed during research projects and is therefore not easily accessible for patients, except for BrainHQ and Luminosity that are two commercial (gaming) companies. As an indication, the price of an annual subscription to these companies is less than USD 100 per year for a full premium account. RehaCom is also already widely used by clinical centers but mostly for research purposes.

This brings us to the second biggest current limitation which is the lack of reimbursement by the social security system. The organization and involvement of healthcare systems in the revalidation process is country-specific and we will not discuss reimbursement specifically here. However, we know that two of the most important barriers to the implementation of telemedicine and telehealth for the patients, regardless of the pathologies or the specialties, are the financial issues and the lack of knowledge and familiarity with the use of (new) technology [77,78]. The pwMS being relatively young, most of them are familiar with smartphones, apps, and mobile technology, therefore the familiarity with the technology should not be an issue for most of the patients [79], but this can be a real barrier for other diseases or patient groups (e.g., older adults with dementia) [80]. Efforts must also be directed to the education of healthcare professionals as they need to be perfectly aware of the technology and its limitations to motivate the patients to use it.

As a result of all the above limitations, in practice, the solutions described in this article are only used by a small fraction of the pwMS. A recent survey performed in the US found that only 3.1% of the pwMS who took part in the survey (*n* = 786) are using mHealth solutions regularly [81].

We will now discuss some ideas for consideration to facilitate the implementation of these solutions in the rehabilitation process.

The first point would be to integrate the mHealth solutions into the healthcare system, with reimbursement for the patients, providing education of the mHealth solutions for healthcare professionals, and the integration of the data collected with the apps (see [54,55,56,57,58,59]) in patients’ medical records. This could speed up and ease the implementation of mHealth in the daily management and rehabilitation of pwMS. This would not only save time but also allow for a more accurate and regular assessment of patients [65]. Furthermore, these assessments could be carried out in the patients’ homes. This fits in perfectly with telemonitoring [82] and the use of real-world data [83]. This would therefore allow the development (by increasing the number of potential users, companies may be more inclined to invest in such solutions) and wider use of these solutions.

A last important point is the sustainability of the studied solutions [84]. The speed of the development of mobile technology (hardware and software) is one of the most important considerations, and the technology becomes quickly obsolete (for example there is a new version of the operating systems [Andoid© or iOS©] on average every 6 months). Thus, the apps that have been developed with the previous version are not supported anymore with the more recent one. This is not much of an issue with the commercial solutions, but it is more problematic with the solutions being developed during research projects.

## 5. Conclusions

This review highlights an important potential of mHealth for pwMS with evidence of a small but significant effect on fatigue and cognition. Although we have seen that current mHealth is, at the moment, not a perfect solution, given the high prevalence of fatigue and cognitive impairment in pwMS and the lack of low-cost tools to assist and stimulate the patients at home, the use of apps could be greatly beneficial.

To develop innovative, effective solutions adapted for pwMS whose cognition, quality of life, functionality, and wellbeing are impaired, researchers, clinicians, policy makers, and app developers will need to further collaborate.

## Figures and Tables

**Figure 1 brainsci-11-01187-f001:**
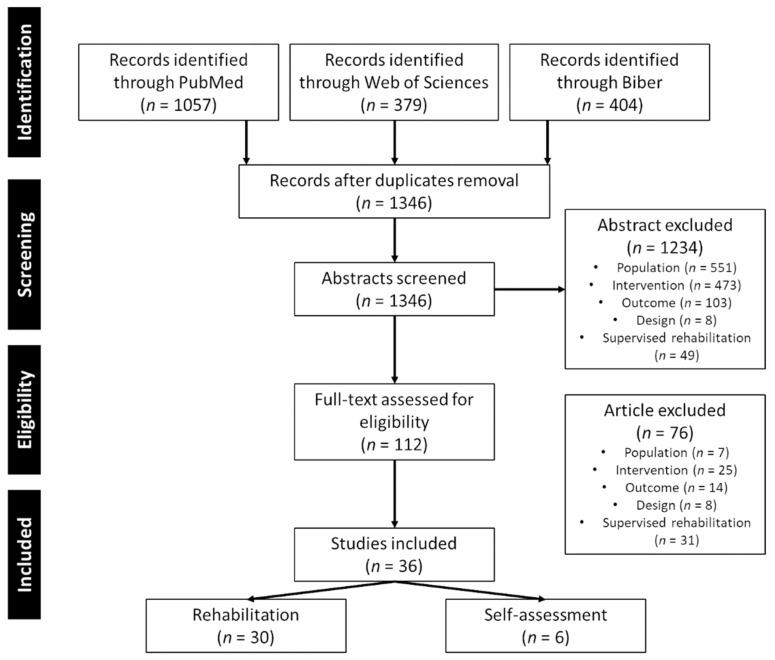
Flowchart of study selection.

**Figure 2 brainsci-11-01187-f002:**
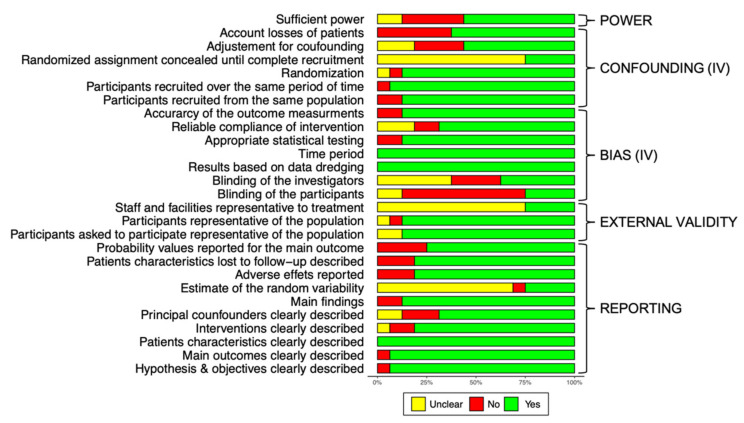
Quality of the study, author’s judgment broken down for each question of the Downs and Black checklist across all included studies (IV): internal validity, for the question about the data dredging the green color indicates that there is no problem and data were acquired directly and have not been imputed.

**Figure 3 brainsci-11-01187-f003:**
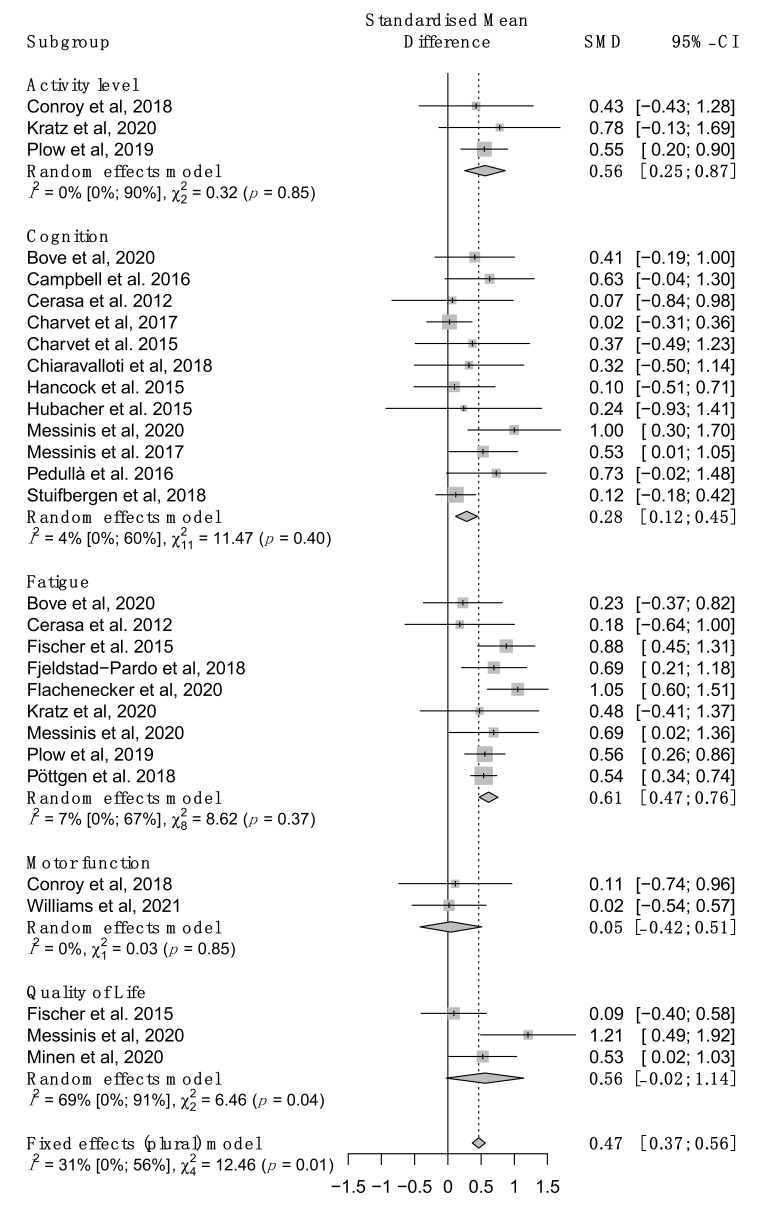
Stratified meta-analysis according to ICF domains, results are indicated with 95% confidence intervals Positive Standardized Mean Differences (SMD) indicates superior efficacy of the mHealth intervention compared to control group [24,26,27,28,29,30,31,33,34,35,36,37,39,40,44,45,47,48,49,51,65].

**Table 1 brainsci-11-01187-t001:** Characteristics of the included studies (study design, mean duration on the intervention, targeted ICF function) on mHealth for rehabilitation and description of the participants.

Study	D&B (/28)	Study Design	Intervention	Duration	Participants	Type of MS and Disability Level	ICF				
							Motor Function	Activity Level	Cognition	Fatigue	Quality of Life
Cerasa et al., 2013 [24]	23	RCT	RehaCom	6 weeks of training (2 × 60 min/week)	17 MS patients 33 (4) years old 85% female	RRMS: 17 EDSS: 3 (0; 4.0)			X	X	
Amato et al., 2014 [25]	21	RCT	Attention Processing Training Program (APT)	12 weeks of training (2 × 60 min/week)	88 MS patients 41 (11) years old 78% female	Type not available EDSS: 2.7 (1.5)			X		
Charvet et al., 2015 [26]	24	RCT	Luminosity	12 weeks of training (5 × 30 min/week)	20 MS patients 40 (8) years old 70% female	RRMS: 20 EDSS: 2 (0; 3.5)			X		
Hancock et al., 2015 [27]	22	RCT	Posit Science inSight (now BrainHQ)	6 weeks of training (6 × 30 min/week)	40 MS patients 50 (6) years old	Type and EDSS not available			X		
Hubacher et al., 2015 [28]	24	RCT	BrainStim	4 weeks of training (4 × 45 min/week)	10 MS patients 46 (7) years old 50% female	RRMS: 10 EDSS: 2 (1.0; 3.5)			X		
Fischer et al., 2015 [29]	23	RCT	Deprexis	9 weeks of training	90 MS patients 45 (12) years old 78% female	RRMS: 40, SPMS: 21, PPMS: 14, unclear: 18					X
Campbell et al., 2016 [30]	22	RCT	RehaCom	6 weeks of training (3 × 45 min/week)	35 MS patients 47 (8) years old 71% female	RRMS: 27, SPMS: 11 EDSS: 5.0 (3.5; 6.0)			X		
Pedullà et al., 2016 [31]	24	RCT	COGNI-TRAcK	8 weeks of training (5 × 30 min/week)	28 MS patients 47 (6) years old 71% female	RRMS: 17, SPMS: 11 EDSS: 3.8 (1.9)			X		
Charvet et al., 2017 [32]	23	RCT	BrainHQ	12 weeks of training (5 × 60 min/week)	135 MS patients 51 (13) years old 77% female	RRMS: 89, SPMS: 35, PPMS: 7, EDSS: 3.5 (2.5; 4.5)			X		
Messinis et al., 2017 [33]	23	RCT	RehaCom	10 weeks of training (2 × 60 min/week)	58 MS patients 46 (10) years old 69% female	RRMS: 58 EDSS: 3.2 (1.0; 5.5)			X	X	
Conroy et al., 2018 [34]	23	RCT	MS HAT system	6 months of intervention Self-paced rehabilitation	54 MS patients 50 (12) years old 77% female	RRMS: 14, SPMS: 35, PPMS: 2 PDSS: 4.1 (1.5)	X	X			
Stuifbergen et al., 2018 [35]	22	RCT	MAPSS-MS	8 weeks of training 2 h/week group session + 3 × 45 min/week home-based training program	183 MS patients 50 (8) years old 87% female	RRMS: 124 EDSS 5.2 (1.6)			X		X
Fjeldstad-Pardo et al., 2018 [36]	21	RCT	CG: exercise sheet tIG: telerehabilitation aIG: in-person rehabilitation + exercise sheet	8 weeks -CG: 5 × week -tIG: 2 × week -aIG: 2 × week	30 MS patients 55 (12) years old 68% female	RRMS: 18, SPMS: 8, PPMS: 4 EDSS: 4.3 (1.1)	X	X	X	X	X
Pöttgen et al., 2018 [37]	23	RCT	ELEVIDA	12 weeks of intervention Self-paced rehabilitation	275 MS patients 41 (11) years old 81% female	RRMS: 200, SPMS: 40, PPMS: 11, unclear: 24				X	X
Cavalera et al., 2019 [38]	24	RCT	MBSR program (mindfulness)-MBI (intervention group) or online psychoeducation (active control group)	8 weeks of training 1 weekly session	121 MS patients 42 (8) years old 34% female	RRMS: 113; SPMS: 8 EDSS: median 3					X
Chiaravalloti et al., 2018 [39]	23	RCT	Processing speed apps (similar to BrainHQ)	5 weeks of training 2/week	21 MS patients 48 (8) years old 75% female	RRMS: 21			X		
Plow et al., 2019 [40]	22	RCT	Contact-control social support intervention Fasting-mimicking diet physical activity plus fatigue self-management intervention PA-only physical activity only intervention	12 week intervention 12 week follow-up Mix between group phone calls and individualized phone calls	208 MS patients 52 (8) years old 85% female	RRMS: 176, SPMS: 11, PPMS: 6, PRMS: 1, unknown: 14		X		X	
Fuchs et al., 2019 [41]	20	Experimental study	BrainHQ	/	51 MS patients 56 years old	RRMS: 35, SPMS: 12, PPMS: 4 EDSS: 4 [2.0; 6.0]			X		
Vilou et al., 2020 [42]	22	Explorative study	BrainHQ	6 weeks of training (2 × 20 min/week) -weekly contact + 2 weeks scheduled visit (semi-assisted)	47 MS patients 35 (16) years old 85% female	RRMS: 47 EDSS: 3.2 (2.0)			X		
Jeong et al., 2020 [43]	23	Retrospective analysis	MS-HAT	6 months of follow-up 2.5 h/week	17 MS patients 60 (11) years old	Type and EDSS not available	X		X		X
Kratz et al., 2020 [44]	24	RCT (pilot)	Web-based and telephone delivered exercises therapy	-Home: 30 min endurance 2× week; 3× week strength training lower extremity + 2 functional exercises per week -in-person: 30 endurance-tr + 30 resistance + home exercise for 8 weeks	20 MS patients 48 (8) years old 90% of female	RRMS: 16, SPMS: 1, PPMS: 1		X		X	
Flachenecker et al., 2020 [45]	23	RCT	Behavior-oriented exercise and physical activity promotion program via web and telephone-based program	12 weeks on intervention -Strength training (1–2 times per week) -Endurance training (10–60 min/1–2 times per week)	64 MS patients 47 (9) years old 62% of female	RRMS: 39, SPMS: 25 EDSS: 4.3 (3.5; 5.0)	X			X	X
Manns et al., 2020 [46]	22	Pre–post intervention (single group)	SitLess+ MoveMore FitBit on (tracking instrument-self monitoring tool) ActivPAL3 (tracking for activity level during 7 days after each time point)	15 weeks of training -7 weeks with SitLess -7 weeks with MoveMore	41 MS patients (39 post intervention and 36 complete follow-up) 50 (10) years old 90% of female	RRMS: 26, SPMS: 11, PPMS: 4 EDSS: 5.5 (3.7)		X		X	
Donkers et al., 2020 [47]	24	RCT (pilot)	Web-based exercise webbasedphysio.com	26 weeks of training Adaptation of the exercises every two weeks	48 MS patients 54 (12) years old 65% of female	Type and EDSS not available	X				X
Messinis et al., 2020 [48]	24	RCT	RehaCom	8 weeks of training (3 × 45 min/week)	36 MS patients 46 (4) years old 66% of female	SPMS: 36 EDSS: 5.5 (4.5; 7.0)			X	X	X
Minen et al., 2020 [49]	23	RCT	RELAXaHEAD	90 days Self-paced training	62 MS patients 40 (10) years old 89% female	Type and EDSS not available					X
Van Geel et al., 2020 [50]	25	Cohort study	Walk-With-Me app	10 weeks of training	12 participants 43 (38.5; 50) years old 100% female	RRMS: 11, SPMS: 1 EDSS not available		X	X	X	X
Bove et al., 2020 [51]	26	RCT	AKL-T03 (web-based)	6 weeks of training (5 × 25 min/weeks)	44 MS patients 51 (13) years old 80% female	RRMS: 33, SPMS: 7, PPMS: 2, CIS: 1, undetermined: 1 EDSS: 3.5 (2.5; 4.5)			X	X	
Tarakci et al., 2021 [52]	24	RCT	Web-based and telphone delivered exercises therapy	12 weeks program (3 × 60 min/week)	30 MS patients 41 (11) years old 77% of female	RRMS: 30 EDSS: 3.4 (1.5)		X		X	X
Williams et al., 2021 [53]	23	RCT	Phone instruction and illustrated training booklet and activity diary	8 weeks of training (2 × 60 min/week)	50 MS patients 51 (10) years old 76% females	RRMS: 31, SPMS: 6, PPMS: 7, undetermined: 6 EDSS not available	X				

The X indicate the main ICF domains assessed. D&B: Downs and Black checklist, RCT: Randomized Controlled Trial, MS: Multiple Sclerosis, RRMS: Relapsing Remitting Multiple Sclerosis, SPMS: Secondary Progressive Multiple Sclerosis, PPMS: Primary Progressive Multiple Sclerosis, CIS: Clinically Isolated Syndrome, EDSS: Expanded Disability Status Scales, PDSS: Patient Determined Disease Steps. MS-HAT: MS Home Automated Telehealth, MAPSS-MS: Memory, Attention, and Problem Solving Skills for Persons with Multiple Sclerosis, MBSR: Mindfulness Based Stress Reduction, PA: Physical Activity.

**Table 2 brainsci-11-01187-t002:** Characteristics of the included studies (study design, mean duration on the intervention, targeted ICF function) on mHealth for self-assessment and description of the participants.

Study	D&B (/28)	Study Design	Intervention	Duration	Participants	Type of MS and Disability Level	ICF				
							Motor Function	Activity Level	Cognition	Fatigue	Quality of Life
Miller et al., 2011 [54]	24	RCT	MCCO-enhanced (Web-Based)	12 months: self-monitoring functioning at any moment, comparing MCCO-original with MCCO-enhanced	206 MS patients	Not available					X
Greiner et al., 2015 [55]	18	Pilot study	MSdialog (Web-Based and App)	6-week study, following stages: 5-min online survey, training teleconference, weekly health reports, 5-min usability survey at weeks 3 and 6, follow-up call interview with selected patients	76 MS patients 68% female	Not available			X	X	
D’Hooghe et al., 2018 [56]	21	Cohort study	MS TeleCoach (Web-Based)	2-week run-in period: assess baseline activity level per patient 12-week period: target number of activity counts gradually increased through telecoaching	75 MS patients 67% female	RRMS: 75 EDSS: 2				X	
Midaglia et al., 2019 [57]	20	Observational study	Floodlight (App)	Active monitoring for 24 weeks: Daily Mood Question: daily, MSIS-29: fortnightly, SDMT: weekly, pinching test: daily, Draw a Shape Test: daily, 5UTT: daily, 2MWT: daily Passive monitoring: gait behavior: continuous, mobility pattern: continuous	101 participants (76 MS patients) 40 years old 70% female	RRMS: 69, SPMS: 4, PPMS: 3 EDSS: 2.4 (1.4)		X	X		
Newland et al., 2019 [58]	18	Pilot study	FatigueApp.com (App)	FatigueApp.com: collect data for 5 weeks on Patient-Reported Outcomes Measurement Information System (PROMIS)	32 MS patients 49 (11) years old 81% female	RRMS: 30, SPMS: 2 EDSS: 3 (2; 4.8)				X	
Pratap et al., 2020 [59]	21	Observational pilot study	ElevateMS (App)	12 weeks Completed baseline assessments, including self-reported physical ability and longitudinal assessments of quality of life and daily health Completed functional tests as an independent assessment of MS-related motor activity	629 participants (490 MS patients) 47 (11) years old 50% female	RRMS: 423, SPMS: 30, PPMS: 42, undetermined: 2		X			X

The X indicate the main ICF domains assessed. D&B: Downs and Black checklist, RCT: Randomized Controlled Trial, MS: Multiple Sclerosis, RRMS: Relapsing Remitting Multiple Sclerosis, SPMS: Secondary Progressive Multiple Sclerosis, PPMS: Primary Progressive Multiple Sclerosis, EDSS: Expanded Disability Status Scales. MCCO-enhanced: The Mellen Center Care Online, MSIS-29: Multiple Sclerosis Impact Scale, SDMT: Symbol Digital Modalities Test, 5UTT: 5 U-Turn Test, 2MWT: 2-Minute Walk Test.

**Table 3 brainsci-11-01187-t003:** Overview of the different mHealth solutions for rehabilitation and self-assessment according to the mean ICF targeted.

Functioning (ICF)	mHealth	
	Rehabilitation	Self-Assessment
Cognition	BrainHQ [27,32,39,41,42] Lumosity [26] RehaCom [24,30,33] BrainStim [28] COGNI-TRAcK [31] MAPPS-MS * [35] APT [25] MS-HAT [43] Walk-With-Me [50] AKL-T03 [51]	MSdialog [55] Floodlight [57]
Fatigue	RehaCom [24,33] ELEVEDIA [37] MAPPS-MS [35] SitLess and MoveMore [46] Walk-With-Me [50] AKL-T03 [51]	MSdialog [55] MS TeleCoach [56] FatigueApp.com [58]
Quality of Life	ELEVEDIA [37] MBSR [38] MS-HAT [43] webbasedphysio.com [47] RehaCom [48] RELAXaHEAD [49] Walk-With-Me [50]	MCCO-enhanced [54] ElevateMS [59]
Activity Level	MS-HAT system [34] SitLess and MoveMore [46] Walk-With-Me [50]	Floodlight [57] ElevateMS [59]
Motor Function	MS-HAT system [34,43] webbasedphysio.com [47]	/

* The cognitive training module of MAPPS-MS is done with Luminosity.

## Data Availability

Not applicable.

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
