# Peer review of "Is mHealth a Useful Tool for Self-Assessment and Rehabilitation of People with Multiple Sclerosis? A Systematic Review"

_brainsci, 2021, doi:10.3390/brainsci11091187_

Round 1

Reviewer 1 Report

the paper is well written, literature search, analysis and synthesis are comprehensive and in line with PRISMA guidelines. The meta analysis compares different tools (mHelth solutions for fatigue, cognition, motor skills), but it is correct and easy to read. The paper cover a field with many new potential perspective .

I just found some minor revisions. I recommend to publish after minor revision.

Best regards

Author Response

Thank you for your interest in this paper, we modified the manuscript according to reviewers’ comments and suggestions.

Reviewer 2 Report

the paper cover an interesting field, with many perspective in the future; it is well written, the literature analysis is comprehensive and detailed, PRISMA guidelines had been followed (even in the absence of a PRISMA checklist).

I just have some minor concerns:

1- introduction at the end of third paragraph "supporting this use, that can be used using mobile technology in pwMS.". This statement needs and English language revision

2- statistical analysis: could you please clarify the method used for SMD  compution

3- 3.4 Description of the available mHealth solutions, MS Dialog paragraph : " to collect and store real-time data regarding administration." ; MS Dialog is a tool designed to better monitor therapeutic adherence in patient taking Rebif. Do you mean interferon administration?

Reviewer 3 Report

In this paper the Authros reviewed different mHealth interventsions for 
rehabilitation and self-assessment of patients with multiple sclerosis (MS). They finally included 36 studies (30 rehabilitation, 6 self-assessment), about 22 different mHealth tools. Overall, mHealth showed small but significant effect on fatigue and cognition.

This is an interesting review addressing an evolving and hot topic in MS care. The paper is well written and the approach to the analysis robust. I have only a few suggestions:

  • A comment about the lack of measures of ecological validity and efficacy of cognitive rehabilitation on everyday-life activities  should be added
  • I suggest to use the word "review" rather than "study" in the abstract and throughout the text

Reviewer 4 Report

Major comments:

  1. For the statistics, the part in methods should be expanded. In results, random-effect models, p-values and forest plots were used for meta analysis but were only very briefly introduced before.
  2. The authors have comprehensively described and discussed the effectiveness based on SMD. A general issue with mHealth training concerns the question of the generalizability and duration of the effects achieved. It would therefore be appropriate to discuss at least the duration of the effect on a meta-level, since there were already significant differences in the time periods in the studies presented.

Minor comments:

  1. Abstract: The Internet starts with a capital letter. At the beginning it should probably read “mobile Internet” instead of “internet mobile”.
  2. It is rather people or persons with multiple sclerosis than “patients with MS”.
  3. Abstract: SMD is not explained within the abstract.
  4. Introduction: As mHealth has a reimbursement issue itself, I would not mention it as a way to overcome reimbursement issues.
  5. Use “health care” or “healthcare” throughout the manuscript, not both.
  6. Methods: What is the rationale to limit the results to studies after 2010?
  7. Results: Stay in past tense, do not start sub types of MS in capital letters.
  8. Results: MCCO only exists as abbreviation. Explain it at first use in the text.
  9. Tables and Figures should be provided with explanations for all used abbreviations. Legends of Table 1-3 and Figure 3 need some additions.
